# A Method for Obtaining the Number of Maize Seedlings Based on the Improved YOLOv4 Lightweight Neural Network

**Jiaxin Gao [1], Feng Tan [1,\*], Jiapeng Cui [2,3] and Bo Ma [4]**

1   College of Electrical and Information, Heilongjiang Bayi Agricultural University, Daqing 163000, China
2   College of Agricultural Engineering, Heilongjiang Bayi Agricultural University, Daqing 163000, China
3   Branch of Suihua, Heilongjiang Academy of Agricultural Mechanization Sciences, Suihua 152054, China
4   Qiqihar Branch of Heilongjiang Academy of Agricultural Sciences, Qiqihar 161006, China
*   Correspondence: bayitf@byau.edu.cn

**Abstract:** Obtaining the number of plants is the key to evaluating the effect of maize mechanical sowing, and is also a reference for subsequent statistics on the number of missing seedlings. When the existing model is used for plant number detection, the recognition accuracy is low, the model parameters are large, and the single recognition area is small. This study proposes a method for detecting the number of maize seedlings based on an improved You Only Look Once version 4 (YOLOv4) lightweight neural network. First, the method uses the improved Ghostnet as the model feature extraction network, and successively introduces the attention mechanism and k-means clustering algorithm into the model, thereby improving the detection accuracy of the number of maize seedlings. Second, using depthwise separable convolutions instead of ordinary convolutions makes the network more lightweight. Finally, the multi-scale feature fusion network structure is improved to further reduce the total number of model parameters, pre-training with transfer learning to obtain the optimal model for prediction on the test set. The experimental results show that the harmonic mean, recall rate, average precision and accuracy rate of the model on all test sets are 0.95%, 94.02%, 97.03% and 96.25%, respectively, the model network parameters are 18.793 M, the model size is 71.690 MB, and frames per second (FPS) is 22.92. The research results show that the model has high recognition accuracy, fast recognition speed, and low model complexity, which can provide technical support for corn management at the seedling stage.

**Keywords:** maize seedlings; detection; YOLOv4; improved Ghostnet; k-means clustering; attention mechanism

## 1. Introduction

Maize is one of the most important crops in Chinese agriculture [1,2]. It is strategically important to ensure the security of grain production, improve stockbreeding as well as the processing industry of grain and oil, and enhance agricultural income and output efficiency. In 2021, China's maize output reached 27255.2 million tons, an increase of 11.8868 million tons, rising 4.56% compared with the output in 2020. The precise number of maize seedlings is able to compute the corresponding emergence rate [3] as well as leakage rate, in favor of timely reseeding and reduction of production loss to the extreme, thus guaranteeing the interests of growers and national food security.

In early times, the number of plants was obtained mainly by fixed (hand-held, tripod) or walking (agricultural machinery, agricultural vehicles, wheelbarrow) equipment, which was single, time-consuming, and laborious. Unmanned Aerial vehicles (UAV) remote sensing [4–12], as a kind of flexible and efficient technology to obtain the information of the farmland environment and crop growth, has been widely applied to agricultural production and scientific research in recent years. Compared with traditional satellite

and aerial remote sensing, UAV has many advantages such as low cost, low loss, repeatability, and low risk, providing a new technical means in order to acquire a large-scale plant number.

In terms of plant number acquisition, researchers mainly extracted the vegetation index [13], color information [14] and plant phenotype [15] through traditional machine vision and image processing technology, and gained different plant numbers according to the obtained characteristics. Jia Honglei et al. [16] converted the grayscale image into a binary image for boundary extraction and image segmentation. They found the geometric center of the corn stalk section and marked it. Liu Shuaibing et al. [17] used the skeleton extraction algorithm and mathematical morphological processes such as deburring to extract the shape of corn seedlings, and to obtain the skeleton of the crop shape. Finally, the Harris corner detection algorithm was used to extract the plant number information of the corn seedling image. Zhao Biquan et al. [18] used the Otsu thresholding method to segment rape plant objects from vegetation index images. The above research methods require a specific shooting environment, angle, and lighting conditions. However, the plant growth environment is complex, and the color and morphological features of the collected images are easily disturbed by various factors, which in turn affects the feature matching effect.

In recent years, with the rapid development of deep learning technology in the agricultural field, the acquisition of plant number information has become more and more convenient, fast, efficient, and accurate. Many scholars at home and abroad have studied the acquisition of plant numbers and achieved results. Chin Nee Vong et al. [19] developed an image-processing workflow based on the deep learning model U-Net for plant segmentation and stand-number estimation. Azam Karami et al. [20] used Red–Green–Blue (RGB) images acquired by drones to identify and count corn through a modified CenterNet. Yang et al. [21] used the improved YOLOv4 model to detect and count wheat ears in the field, and added the Convolutional Block Attention Module (CBAM) module to the YOLOv4 network to enhance the feature extraction ability of the network. Guo Rui et al. [22] detected the number of soybean pods in the YOLOv4 model by integrating the k-means clustering algorithm and the improved attention mechanism module. The results show that the model had strong generalization ability. Zhang Hongming et al. [23] proposed a seedling acquisition detection model (FE-YOLO) based on the feature enhancement mechanism, which reduced the complexity of the network and realized the rapid acquisition of the number of maize seedlings. All the above methods have made certain contributions to the acquisition of plant quantity information. However, the model detection accuracy is low and the number of model parameters is large. Training takes a long time, making it difficult to arrange in mobile terminals and embedded chips.

This study innovatively proposed an improved lightweight target detection neural network for maize seedling quantity acquisition. In this method, the YOLOv4 network is used as the backbone model. Firstly, the improved Ghostnet network and depthwise separable convolution are introduced to reduce the number of model parameters. Then, introducing the CBAM attention module and the k-means clustering algorithm improves the model's ability to recognize tiny targets. Finally, on the premise of comprehensively analyzing the size of the prior boxes and the target size of the maize seedling to be detected, the detection branch used to detect large-sized targets is selected to be removed, and an improved lightweight target detection neural network is finally obtained. The improved lightweight object detection neural network improves the model's training efficiency and recognition performance by transferring public knowledge pre-trained on the COCO dataset. The experimental analysis shows that this method has the potential for mobile deployment and is practical and effective.

## 2. Materials and Methods

### 2.1. Overview of Experimental Area

The experimental area is located in the modern Agricultural Science and Technology Demonstration Park, Duerbote Mongolian Autonomous County, Daqing City, Heilongjiang Province, with the central geographic coordinates of (46°49′ N, 124°26′ E). The altitude is about 147 m. The planting crop is corn. The area has a large temperature range between day and night, abundant with sufficient solar and wind energy resources, which is appropriate for crop planting.

### 2.2. Image Acquisition

The UAV image test data were collected from 10:00 to 13:00 on 12 June 2021 in the Agricultural Park of Duerbert Mongolian Autonomous County. In this experiment, maize is in V3 growth stage. In the acquisition of UAV images, to avoid the loss of texture feature information of some images due to cloud occlusion, weather conditions with stable solar radiation intensity and a clear and cloudless sky were selected for acquisition. In the process of data acquisition, Pix4Dcapture was used for route planning, and the UAV route overlap degree was set as 80%, side overlap degree as 70%, and flight altitude as 10 m for the experiment. The ground sampling distance was 0.44 pixel/cm. DJI Phantom 3 Professional UAV is a remote sensing platform with a total mass of 1280 g and a maximum horizontal flight speed of 57.6 km/h in a windless environment. The battery is LiPo (4 S, 4480 mAh). The UAV is equipped with a 1/2.3 inch CMOS image sensor, effective pixels of 12.4 million, and FOV94 ° 20 mm f/2.8 lens. The data source collected is image data in *.jpg format.

### 2.3. Data Set Construction and Preprocessing

2.3.1. Image Preprocessing

After the aerial image is exported from the UAV storage device, the Pix4DMapper [24] software is used to generate the mosaic image. First, feature points are automatically calculated in each image. Then, the feature points are automatically matched in the image sequence, which needs to be repeated many times. After the matching is completed, enter the stage of generating the dense point cloud. Finally, a Mosaic image is generated as shown in Figure 1b.

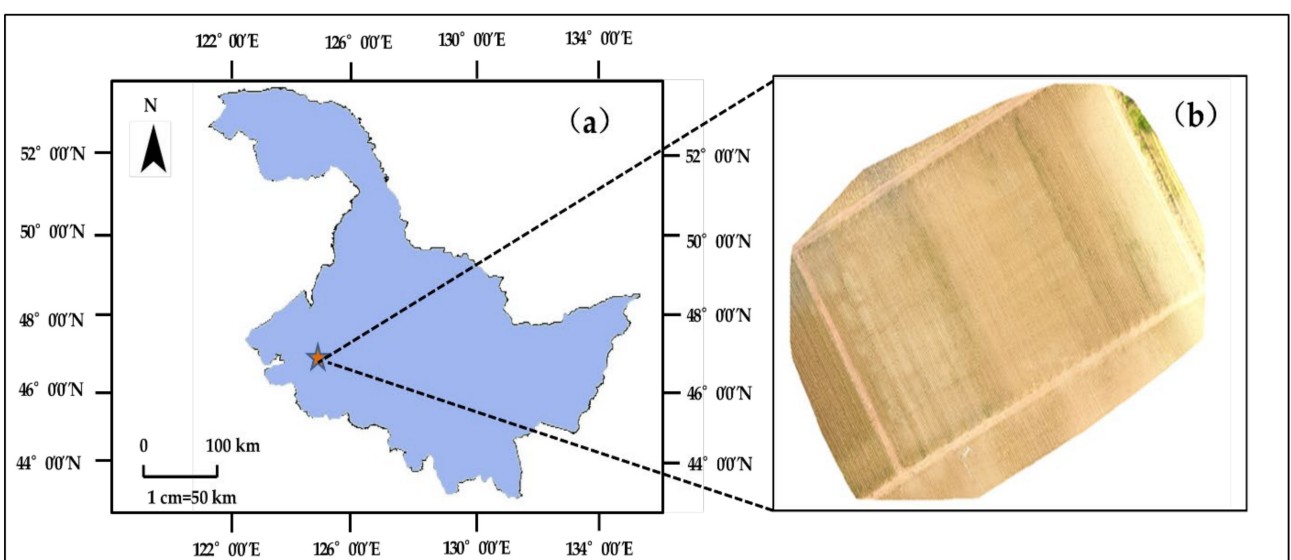

**Figure 1.** Study area: (**a**) experimental-area location; (**b**) splicing diagram of test field.

2.3.2. Data Set Construction

A total of 300 [25–27] images of maize seedlings were collected in this experiment. Considering the large size of the collected single image, dense seedlings, and small pixel area of the whole image, sample production will confine itself to a large amount of sample area, and the image cannot be directly scaled and detected, affecting the speed of network detection. Therefore, the Python language is used in the research to slice the collected images to improve the speed of network training and detection.

The sliced images were screened to eliminate fuzzy and distorted images, and 500 images were randomly selected as data sets. The data sets were divided into a 9:1 ratio, 90% divided into training set and verification set according to 9:1, and the remaining 10% were used as test sets. Finally, the sample numbers of the training set, validation set, and test set were 405, 45, and 50. The LabelImg tool was used to annotate the data of corn seedlings in the image, and the .XML file including ground truth was obtained after the annotation, and the data set was constructed according to the Pascal Visual Object Classes (VOC) [28] data format.

*2.4. YOLOv4 Network Model*

Target detection algorithms based on deep learning are mainly divided into two categories. One is a two-stage target detection method represented by RCNN [29], SPP Net [30], Fast RCNN [31], and Faster RCNN [32], whose main idea is to generate regions first and then conduct classification and recognition through a convolutional neural network. The other is the single-stage target detection method represented by the Single Shot MultiBox Detector (SSD) [33], You Only Look Once (YOLO), etc. Its main idea is to predict object classification and location by extracting features directly from the network without using a region proposal. As a classical single-stage target detection network, YOLO has fast inference speed and high accuracy, among which YOLOv4 [34] makes a series of improvements based on YOLOv3 [35], greatly improving its speed and accuracy. YOLOv4 is mainly composed of four parts: input terminal, backbone network, neck network and head network. The Mosaic data augmentation method is designed, and the input images are merged by random clipping, scaling, and spatial arrangement. At the same time, training techniques, such as the learning rate cosine annealing attenuation method, are used. This method not only enriches the data set but also improves the training speed of the network. The backbone network is the CSPDarknet53 network, which includes five Cross Stage Partial (CSP) modules that first divide the feature mapping of the base layer into two parts and then combine them through a cross-stage hierarchy. The Mish activation function, which is smoother than the Leaky ReLU activation function, can further improve the accuracy of the model. The Mish function [36] expression is:

$$\text{Mish} = x \times \tan h(\ln(1 + e^{x})) \tag{1}$$

where x is the input value, tanh is the hyperbolic tangent function, and ln is the logarithmic function of the number based on e.

The Spatial Pyramid Pooling (SPP) structure, using maxpooling kernels such as $\{1 \times 1, 5 \times 5, 9 \times 9, 13 \times 13\}$, stitches together feature maps of different scales. Compared with simply using the $k \times k$ maxpooling, it can more effectively increase the receiving range of backbone features and significantly separate the most critical context features. The Path Aggregation Network (PANet) adds a bottom-up path augmentation structure after the top-down feature pyramid, which contains two PAN structures, and the PAN structure is modified. The original PAN structure uses a shortcut connection to fuse the down-sampled feature map with the deep feature map, and the number of channels of the output feature map remains unchanged. The modified PAN uses the concat operation to connect the two input feature maps, and merge the channel numbers of the two feature maps. The top-down feature pyramid structure conveys strong semantic features, and the bottom-up path augmentation structure makes full use of shallow features to convey strong

positioning features. PANet can make full use of shallow features, and for different detector levels, feature fusion of different backbone layers to further improve feature extraction capabilities and improve detector performance.

The head network adopted multiple l x 1 alternately and the size of a 3 × 3 convolution kernels for the convolution operation, being able to experiment with the size of the figure, and then forecast the figure; the characteristics of each layer will generate the corresponding three-box criterion to predict the box, whether the forecast box containing the features related to the information, and detect the target. Finally, the final prediction box is obtained by the non-maximum algorithm and the set prior box.

### 2.5. Improved YOLOv4 Network Model Design

#### 2.5.1. Improved Ghostnet Feature Extraction Network

The backbone network of YOLOv4 is CSPDarkNet53. Although the feature extraction capability of the CSPDarkNet53 backbone network is strong, the calculation is complex and requires more memory space. This study proposes an improved YOLOv4 lightweight neural network model. Based on the original YOLOv4 network, the improved Ghostnet network is used as the backbone feature extraction network. Ghostnet [37] is a novel end-to-side neural network architecture proposed by Huawei's Noah's Ark Lab. MobileNet and ShuffleNet [38] introduce depthwise convolution and shuffling operations, using smaller convolution kernels (floating point operations) to build efficient CNN, but the remaining 1 × 1 convolutional layers will still take up a lot of memory. While Ghostnet maintains a similar recognition performance, the total number of parameters required by the model and the computational complexity are reduced. The ghost module divides the ordinary convolution into two parts. First, an ordinary 1 × 1 convolution is performed to generate the feature concentration of the input feature layer. Next, depthwise separable convolutions are used to obtain similar feature maps with feature enrichment. Ghost bottlenecks consist of ghost module; Ghostnet contains two bottleneck structures, stride = 1 for feature extraction and stride = 2 to reduce the number of channels. Ghost bottlenecks consist of two parts, the backbone part and the residual edge part. The structure is shown in Figure 2 below.

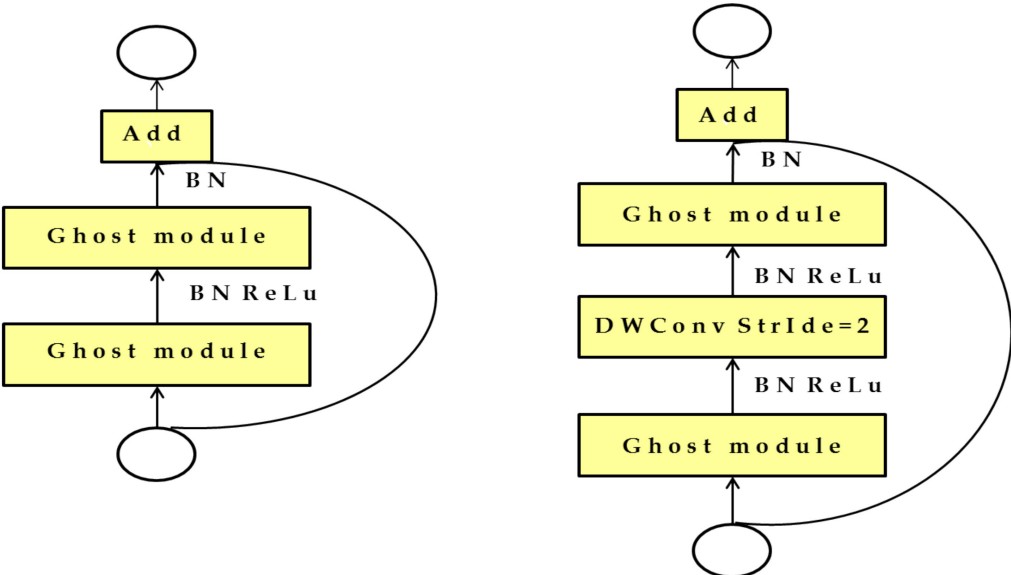

**Figure 2.** Structure diagram of ghost bottlenecks.

The entire Ghostnet is composed of Ghost bottlenecks; after the stacking of Ghost bottlenecks, the feature layer of $7 \times 7 \times 160$ is finally obtained. Use a $1 \times 1$ convolution block to adjust the number of channels to obtain a $7 \times 7 \times 960$ feature layer. Through a global average pooling at the tail, the size of the feature map is reduced from $7 \times 7$ to $1 \times 1$. After using a $1 \times 1$ convolution block to adjust the number of channels, a $1 \times 1 \times 1280$ feature layer is obtained.

This study adopts the improved ghost module. In the feature maps extracted by mainstream deep neural networks, rich and even redundant information usually ensures a comprehensive understanding of the input data. Therefore, in the improved Ghost module, the input image is first subjected to a $1 \times 1$ ordinary convolution to increase the dimension and define the dimension-raising coefficient T = 6. Ensure that sufficient information is extracted and increase the interaction of cross-channel information. Next, in the ghost module trunk branch, first use depthwise convolution. After the dimension increase, use cheap operations to get more redundant feature maps on the input data. After passing through the $1 \times 1$ convolution layer, these redundant features are concentrated in the form of reverse residuals. In this way, more feature information can be obtained with less computational cost. In the residual branch of the ghost module, use $1 \times 1$ convolution to adjust the channel dimension of the input data to facilitate the feature concat combination with valid features extracted from the trunk branch. Then, combine the different feature maps together to form a new output. The Ghostnet network can use the SENet module to achieve feature enhancement of the channel attention. However, due to the lack of fine-grained feature improvement, this module has a large amount of computation and a limited feature improvement effect. Therefore, in this study, the SENet module is removed to further reduce the number of model parameters. The structure diagram of the improved ghost module is shown in Figure 3.

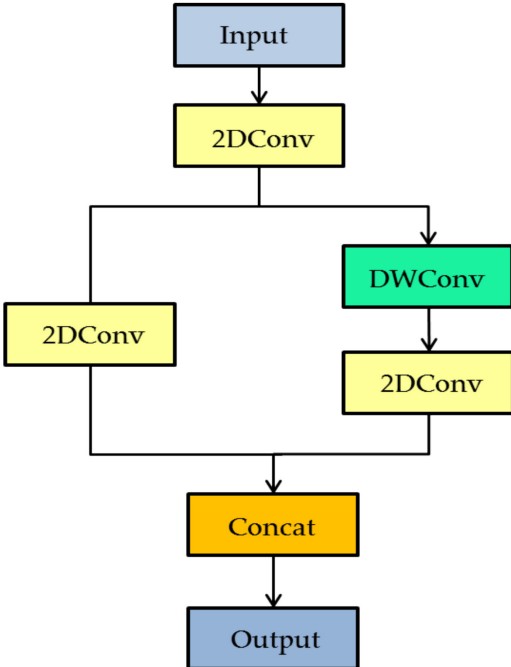

**Figure 3.** Improved ghost module.

### 2.5.2. Depthwise Separable Convolution

Depthwise Separable Convolution [39] is divided into depthwise convolution and pointwise convolution. Depthwise Separable Convolutions consider channels and spatial regions separately. Dividing ordinary convolution into two processes for operation can learn rich feature information with fewer parameters. Depthwise convolution and pointwise convolution are shown in the Figures 4 and 5 below. In order to further reduce the number of parameters of the network, depthwise separable convolution was introduced in the PANet and YOLO Head structures to replace the standard convolution in the original network. The formula for calculating the ratio of depthwise separable convolution to traditional convolution is:

$$\frac{K^2M + MO}{K^2MO} = \frac{1}{O} + \frac{1}{K^2} \tag{2}$$

where K is the convolution kernel size of the depthwise separable convolution, M is the number of input feature maps, and O is the number of output feature maps. The computational cost of depthwise separable convolution is about $1/K^2$ of that of standard convolution.

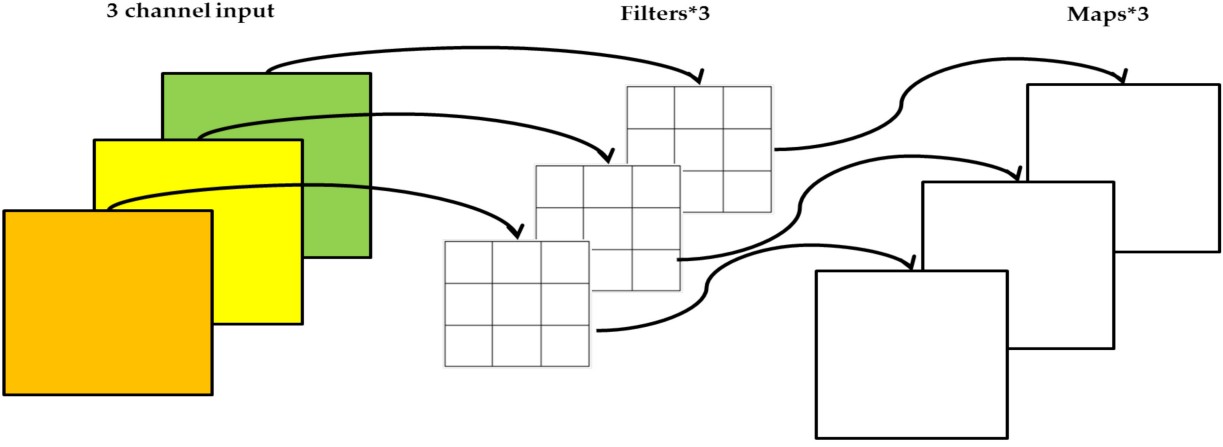

**Figure 4.** Depthwise convolution.

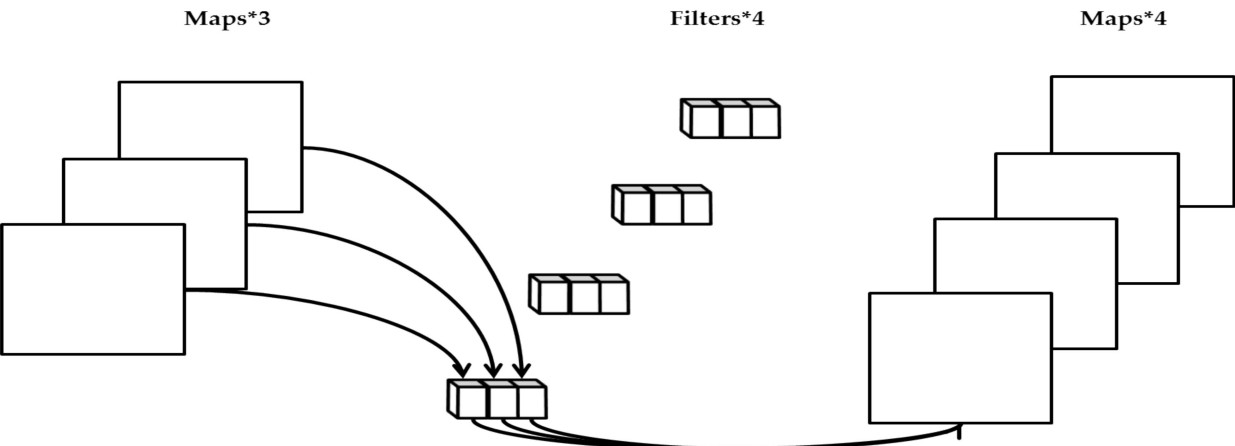

**Figure 5.** Pointwise convolution.

### 2.5.3. CBAM Attention Mechanism

CBAM [40] is an efficient and lightweight attention module. This module can be integrated into any convolutional neural network structure and trained end-to-end with the base network. It contains two independent sub-modules, the Channel Attention Module (CAM) and the Spatial Attention Module (SAM). The two modules are combined in the way of the first channel attention module and then the spatial attention module. Compared with

SENet [41], CBAM adds a spatial attention module, since in this study, the maize seedlings, after generating orthophotos, have good positional characteristics in the two-dimensional plane. Therefore, the CBAM module is selected to assign the weights of the channel features and spatial features of the feature map, increase the weights of useful features, and suppress the weights of invalid features. This module can make the network pay more attention to the target area with important information, obtain relevant information, and improve the overall accuracy of target detection. The CBAM module is shown in Figure 6. After many trials, the module works best when placed at the front of the YOLOv4 neck network. Therefore, the improved YOLOv4 lightweight neural network in this study is trained and predicted based on this.

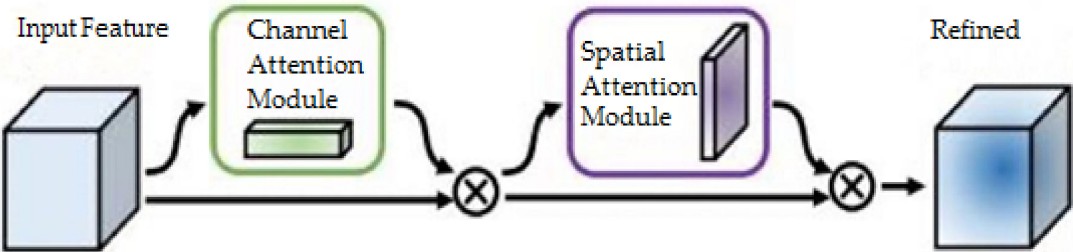

**Figure 6.** Structure of CBAM.

### 2.5.4. K-means Clustering Adjusts the Target Prior Box

The original YOLOv4 prior box is based on the COCO dataset. Including 80 categories, the size of the a priori box is different for different categories. In order to make it more suitable for the detection of maize seedlings, this study adjusted the prior box in the original YOLOv4 model using the k-means clustering algorithm. Randomly select k boxes as the cluster center, and calculate the distance between all other anchor boxes and the cluster center. The calculation of distance is based on the concept of Intersection Over Union (IOU). All anchor boxes are divided into k regions according to the distance, and the distance of these k regions is averaged. The average value is taken as the cluster center of the region again, and the iteration stops until the cluster center does not change. The formula for the distance between the cluster center and other anchor boxes is as follows:

$$\text{distance}(a, b) = 1 - \frac{\min(w_1, w_2) \times \min(h_1, h_2)}{w_1 h_1 + w_2 h_2 - \min(w_1, w_2) \times \min(h_1, h_2)} \qquad (3)$$

In the formula, a is the anchor box of other tags, b is the anchor box position of the cluster center traversing all the tag files, w1 is the width of the cluster center, h1 is the height of the cluster center, w2 is the width of other tag anchor boxes, h2 is the height of the anchor box for other labels. We calculate the size, width, and height of the anchor box, and use the obtained data to perform k-means clustering. The anchor box coordinate information of all training data is used as the input sample, and the size of the 9 cluster center a priori boxes is finally generated by iteration as (18, 14), (46, 23), (29, 44), (58, 41), (89,33), (46,65), (85,55), (71,85), (128,73).

### 2.5.5. Improved Multi-Scale Feature Fusion Network Structure

YOLOv4 uses different detection layers to detect objects of different sizes. For a 416 × 416 input image, the sizes of the three detection layers are 13 × 13, 26 × 26, 52 × 52. The feature maps of the three detection layers are down-sampled 8 times, 16 times, and 32 times, respectively. Each detection layer corresponds to 3 a priori boxes, containing a total of 9 a priori boxes. The smaller the size of the feature map, the larger the corresponding a priori box size. The largeness of the area corresponding to each grid cell in the input image is generally responsible for detecting objects with larger sizes. On the contrary, the larger the size of the feature map, the smaller the corresponding a priori box size, the smaller the area corresponding to each grid cell in the input image, generally responsible

for detecting objects with smaller sizes. Therefore, the 13 × 13 detection layer is suitable for the detection of large objects and 52 × 52 detection layers are suitable for small target detection. After k-means clustering, it is found that the image size of maize seedlings is relatively uniform and small in size. Therefore, this study removes the 13 × 13 detection layer used to detect large objects and the prior boxes are changed from the original 9 to 6. This improved method compresses model parameters while ensuring model detection accuracy. The improved network structure is shown in Figure 7.

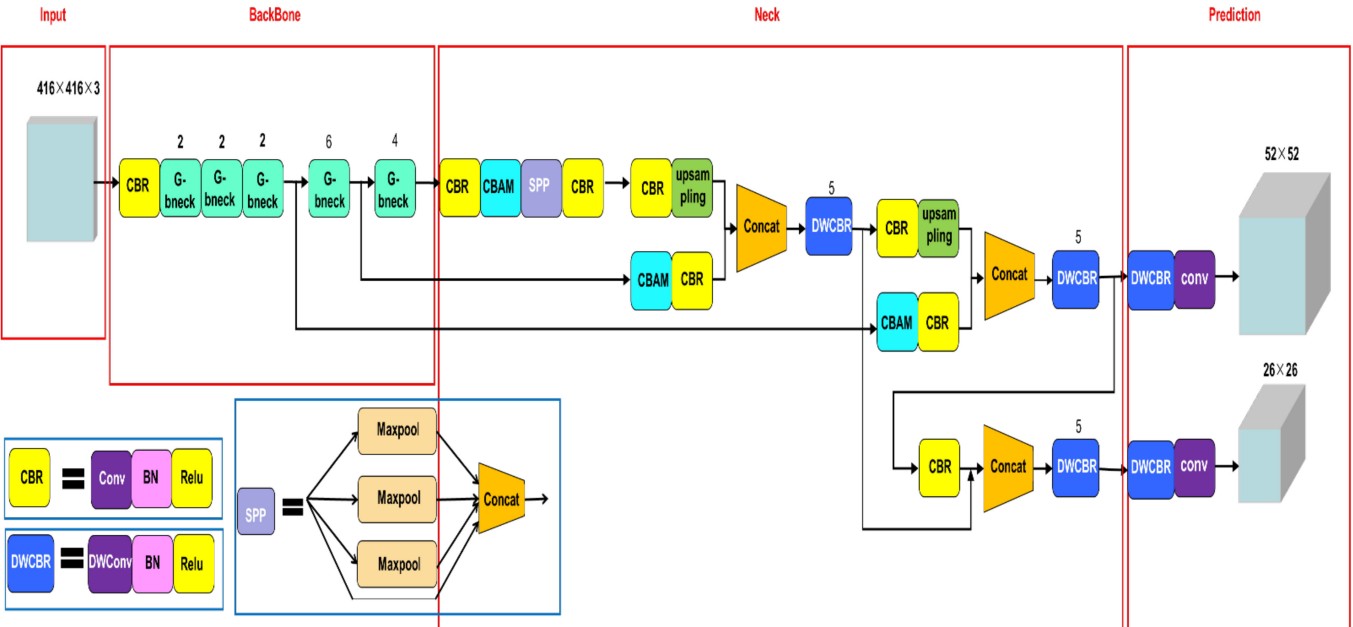

**Figure 7.** Improved multi-scale feature fusion network structure.

### 2.5.6. Pre-training the Original Network with Transfer Learning

Transfer learning [42] refers to the idea that knowledge learned in one environment is used in another domain to improve its generalization performance. With the development of deep learning, the network model gradually increases from the first few layers. The hardware requirements for model training are becoming higher and higher, and the training time is longer. Transfer learning can help us solve this problem. We load the weight parameters trained by others into our model, which can quickly train an ideal result without requiring a large dataset.

In order to improve the training speed, the data set in this study is limited. We use the idea of transfer learning to transfer the simple edge information learned by the Ghostnet-YOLOv4 model in the COCO dataset to the maize seedling detection network. Use initial weights to pre-train the improved YOLOv4 lightweight neural network model; first, freeze some convolutional layers. When the model gradually converges, unfreeze the entire network model and train the entire network. The whole process of detecting the number of maize seedlings is shown in Figure 8.

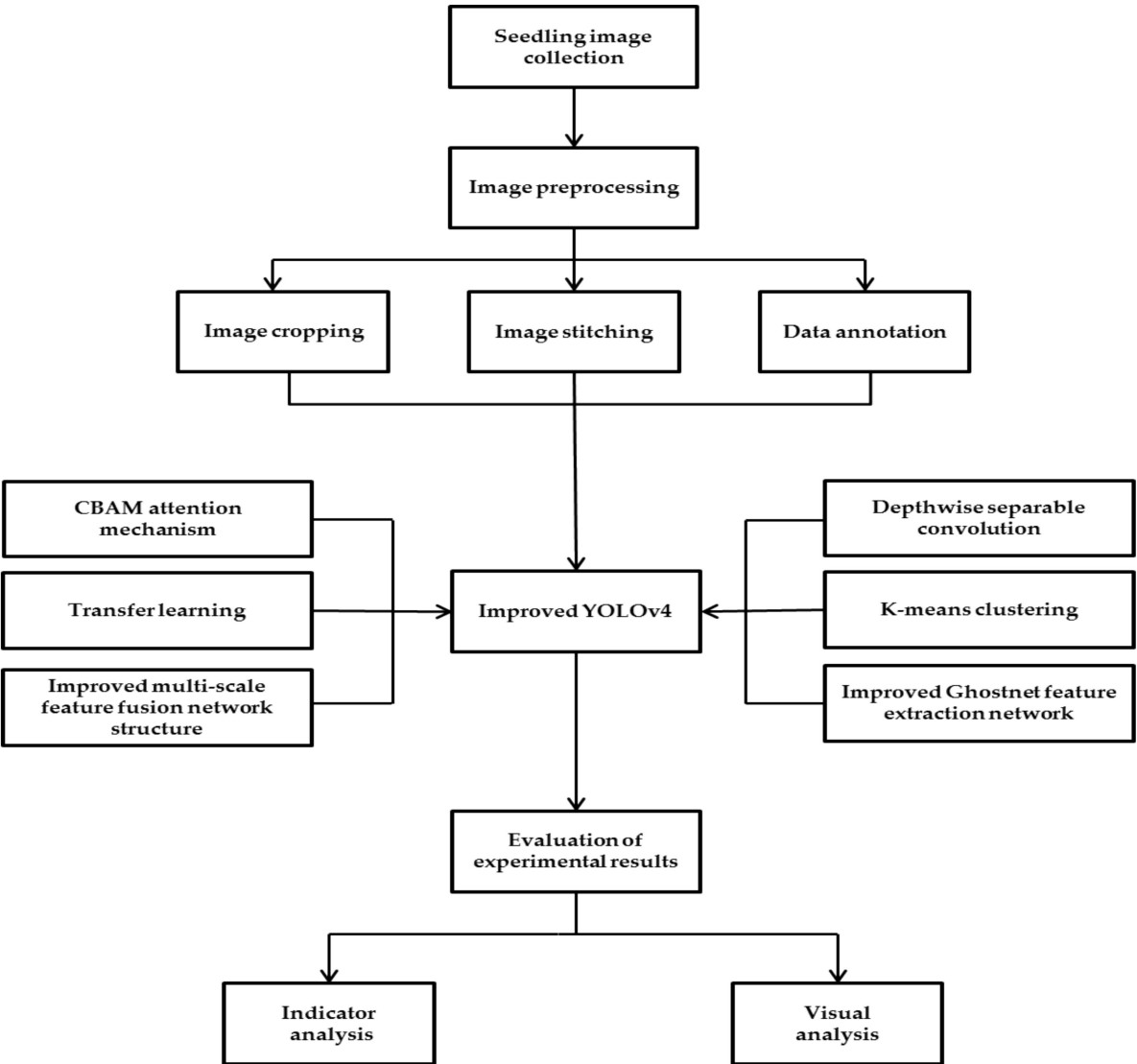

**Figure 8.** Flow chart of seedling number detection.

*2.6. Test Evaluation Index*

To objectively measure the target detection effect of the model on maize seedlings, the Precision (P), Recall (R), Harmonic Average F1 Value (F1), Average Precision (AP), Mean Average Precision (mAP), the number of network parameters, the model size, and FPS were used to evaluate the trained model. The IOU value was 0.5 in the experiment. The calculation formulas of P, R, F1 are shown in the following formulas.

$$\text{Precision} = \frac{\text{TP}}{\text{TP} + \text{FP}} \times 100\% \tag{4}$$

$$\text{Recall} = \frac{\text{TP}}{\text{TP} + \text{FN}} \times 100\% \tag{5}$$

$$\text{F1} = \frac{2 \times \text{Precision} \times \text{Recall}}{\text{Precision} + \text{Recall}} \tag{6}$$

Among them, True Positive (TP) represents the number of correctly detected maize seedlings; False Positive (FP) represents the number of misclassified maize seedlings; False Negative (FN) represents the number of missed maize seedlings; F1 represents the harmonic average of accuracy and recall. When F1 is closer to 1, the model is better optimized. AP represents the area composed of the PR curve and the coordinate axis. The higher the

AP value is, the better the performance of the target detection algorithm is. The mAP represents the AP average of multiple categories, and its value represents the general detection performance of the algorithm for different categories. Since this study only performed single-class target detection for maize seedlings, the mAP and AP values were the same, and both were the area of the PR curve of maize seedlings. The specific definition is shown in Equation (7).

$$mAP = \frac{1}{N} \sum_{m=1}^{N} \int_{0}^{1} PRdr \qquad (7)$$

In the formula, N is the number of categories, and to compare the model complexity, the model size, the number of network parameters, and the number of frames per second are used as model complexity evaluation parameters.

## 3. Results and Analysis

### 3.1. Test Platform and Training Parameters Setting

This experiment is based on the PyTorch framework, and the experimental environment is shown in Table 1. The model accuracy of different backbone feature extraction networks and the differently modified parts of the improved model are compared on the Windows operating system. Verify the model performance on the same validation set. In this experiment, the input image pixels are 416 × 416. The training was divided into two stages, and the whole stage was trained for 2000 epochs. For the first half of the stage, the backbone feature extraction network of the model was trained in 500 epochs by freezing. The initial value of the learning rate was set to $1 \times 10^{-3}$, and the batch size was set to 16. The learning rate adjustment is realized by the cosine annealing decay algorithm, and the decay coefficient is 0.0005. For the second half of the stage, the backbone feature extraction network was unfrozen, and the entire model was further trained for 1500 epochs with an initial learning rate of $1 \times 10^{-3}$, and the batch size was set to 8. Save the weight file every 200 generations of training (epoch) on the training set and generate a log file to output the loss value of the training set and the validation set. The loss value curves of the training set and validation set of the improved model in this paper are shown in Figure 9.

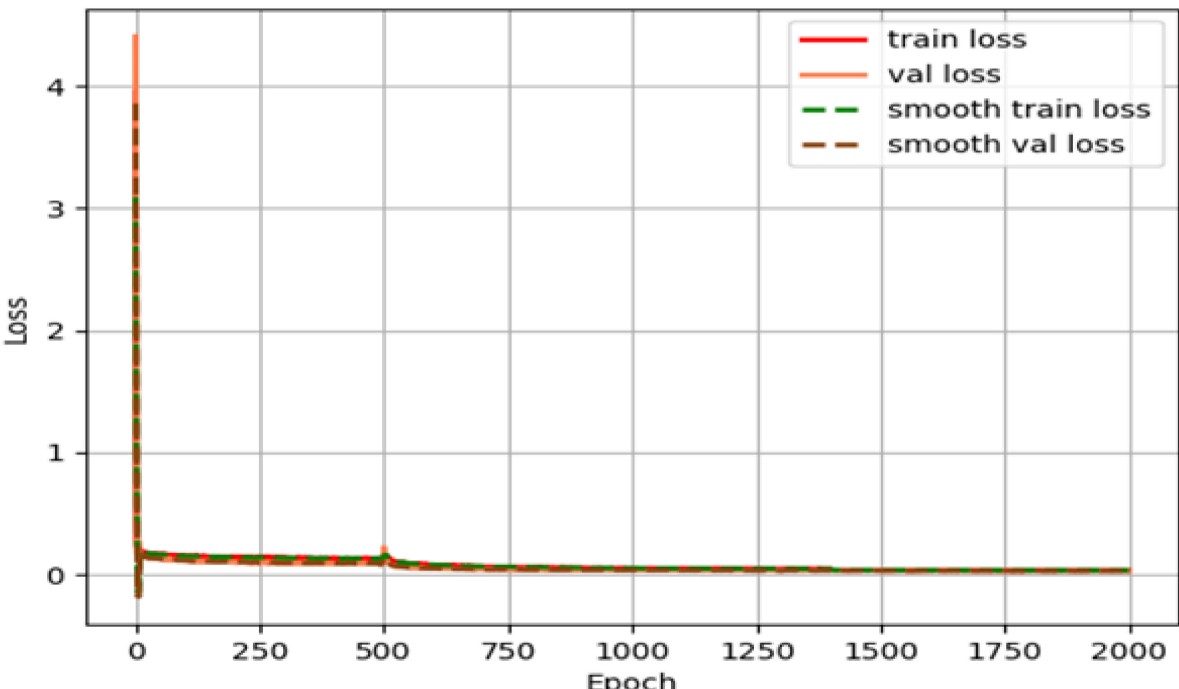

**Figure 9.** The curve of the loss value changing with the number of iterations.

**Table 1.** Test environment.

| Configure | Parameter |
|---|---|
| CPU | Intel(R) Xeon(R) CPU E5-2680 v4 @ 2.40 GHz |
| GPU | NVIDIA GeForce RTX 3060 16 G |
| Operating System | Windows10 |
| Acceleration Environment | Cuda 11.3 |
| Development Platform | PyCharm |
| Others | Numpy1.17.0 Opencv4.1.0 |

*3.2. Comparison of Seedling Test Results*

3.2.1. Comparison of Different Backbone Feature Extraction Networks

It is convenient to place the model on the mobile side to reduce the number of model parameters and design an improved YOLOv4 lightweight neural network using improved Ghostnet as the backbone network. In order to verify the rationality of the designed improved YOLOv4, a comparison experiment was conducted on the test set to compare different backbone feature extraction network models. The mAP, precision, recall, F1-score, model size, number of network parameters, and FPS of different models were obtained, as shown in Table 2.

**Table 2.** Comparison of different backbone feature extraction networks.

| Model | P (%) | R (%) | mAP (%) | F1 | Model Size (MB) | Parameters (M) | FPS |
|---|---|---|---|---|---|---|---|
| YOLOv4 | 90.87 | 74.06 | 88.35 | 0.82 | 243.90 | 63.938 | 19.21 |
| Vgg-YOLOv4 | 92.65 | 90.87 | 94.32 | 0.92 | 197.50 | 51.773 | 26.18 |
| Densenet121-YOLOv4 | 94.16 | 87.74 | 95.10 | 0.94 | 168.89 | 44.274 | 23.89 |
| Mobilenetv1-YOLOv4 | 94.04 | 89.06 | 94.54 | 0.91 | 154.60 | 40.527 | 29.46 |
| Mobilenetv3-YOLOv4 | 93.66 | 90.62 | 94.87 | 0.92 | 150.93 | 39.565 | 28.90 |
| Ghostnet-YOLOv4 | 93.24 | 89.54 | 94.07 | 0.91 | 149.78 | 39.264 | 32.77 |

As seen from the table, compared with the original YOLOv4 model, this study uses Ghostnet as the backbone network, and the mAP, precision, recall, and F1 are greatly improved. The number of network parameters and model size is greatly reduced, and the FPS is improved. Through experiments, it is found that although Ghostnet-YOLOv4 is close to other models in terms of various accuracy indexes, the number of network parameters and model size of Ghostnet-YOLOV4 is the smallest among the following models and the largest in FPS among the following models. All indicators of Vgg-YOLOV4 and Densenet121-YOLOV4 have a good performance, but the number of network parameters and model size is larger, and the FPS decreases more than that of Ghostnet-YOLOV4. We use Ghostnet as the backbone network, and the model has improved various indicators to varying degrees. Among them, the mAP has increased by 5.72%, the precision rate has increased by 2.37%, the F1 has increased by 0.09, the recall rate has increased by 15.48%, the number of network parameters decreased by 38.59%, the model size decreased by 94.12 MB, and the FPS reached 32.77. Therefore, the Ghostnet-YOLOV4 lightweight neural network improves precision and has apparent advantages in mobile devices and embedded applications. Therefore, this study further improves the model detection effect and reduces the model size.

3.2.2. Comparison of Test Results of Different Improved Structures

The original YOLOv4 algorithm has a large number of model parameters, which is not conducive to placement on the mobile terminal. There are false detections and missed detections for dense and overlapping targets. Therefore, an improved YOLOv4 lightweight neural network is proposed. The specific improvement methods are as follows: ①Improved multi-scale feature fusion network structure. By analyzing the detection targets in this study, it is found that the size of maize seedlings is relatively uniform, and the target box

is small. Therefore, this study removes the 13 × 13 detection layer for detecting large objects to reduce the number of network parameters. ② The improved Ghostnet is used as the backbone feature extraction network. ③ Depthwise separable convolution. In order to reduce the number of model parameters as much as possible, this study replaces the standard convolution in PANet and YOLO head with the depthwise separable convolution. ④ K-means clustering. The anchor boxes of the original YOLOv4 network are based on the COCO dataset, and this paper re-clusters the anchor boxes to make them more suitable for the detection of maize seedlings. ⑤ CBAM attention mechanism. In order to verify the superiority of lightweight YOLOv4 improvement, we conducted comparative experiments on the test set. When IOU = 0.5, the test results are shown in Table 3.

**Table 3.** Comparison of detection results of different improved structures.

| Model | P (%) | R (%) | mAP (%) | F1 | Model Size (MB) | Parameters (M) | FPS |
|---|---|---|---|---|---|---|---|
| YOLOv4 | 90.87 | 74.06 | 88.35 | 0.82 | 243.90 | 63.938 | 19.21 |
| YOLOv4 + ① | 92.12 | 77.50 | 90.75 | 0.84 | 179.29 | 47.001 | 31.94 |
| YOLOv4 + ① + ② | 93.61 | 87.98 | 94.25 | 0.91 | 127.49 | 33.422 | 26.89 |
| YOLOv4 + ① + ② + ③ | 94.69 | 90.02 | 94.93 | 0.92 | 71.61 | 18.772 | 24.89 |
| YOLOv4 + ① + ② + ③ + ④ | 97.28 | 91.82 | 96.91 | 0.94 | 71.61 | 18.772 | 24.87 |
| YOLOv4 + ① + ② + ③ + ④ + ⑤ | 96.25 | 94.02 | 97.03 | 0.95 | 71.69 | 18.793 | 22.92 |

Note: ① Improved multi-scale feature fusion network structure; ② Improved Ghostnet backbone feature extraction network; ③ Depthwise separable convolution; ④ K-means clustering; ⑤ CBAM attention mechanism.

On the basis of YOLOv4, the improved multi-scale feature fusion network structure improves the recall rate by 3.44%, mAP by 2.40%, F1 value by 0.02, precision by 1.25%, network parameters reduced by 26.49%, model size decreased by 64.61 MB, and FPS increased by 12.73 compared to the original YOLOv4. The experimental results show that the improved multi-scale feature fusion network structure ensures that the network recall, mAP, F1 value, and precision are higher while reducing a certain amount of model parameters, and the FPS has been significantly improved.

In the network that replaced the backbone feature extraction network and improved multi-scale feature fusion structure, compared with the original YOLOv4 network, the recall rate increased by 13.92%, mAP increased by 5.90%, F1 value increased by 0.09, precision increased by 2.74%, network parameters are reduced by 30.516 M, the model size is reduced by 116.41 MB, and the FPS is increased by 7.68. Through comparison, it can be seen that the indicators of the model have been greatly improved, and the number of network parameters has significantly been reduced. It shows that this study's improved Ghostnet feature-extraction network can extract the feature information of maize seedlings well. Compared with the network that only improves the multi-scale feature fusion network structure, the recall rate is increased by 10.48%, the mAP is increased by 3.5%, the F1 value is increased by 0.07, the precision is increased by 1.49%, network parameters are reduced by 13.579 M, and the model size is reduced by 51.80 MB, but the FPS is reduced by 5.05. It shows that the replacement of the backbone network can improve the detection accuracy of the model, but the FPS decreases. The validity of this partial improvement in the field of maize seedling detection was confirmed.

Compared with the original YOLOv4 network, after adding the depthwise separable convolution to the model, the recall rate is increased by 15.96%, the mAP is increased by 6.58%, the F1 value is increased by 0.1, the precision is increased by 3.82%, and the network parameters are reduced by 45.166 M, the model size is reduced by 172.29 MB, and the FPS is increased by 5.68. Compared with the network that replaced the backbone feature extraction network and improved multi-scale feature fusion structure, the recall rate has improved greatly, and the improvement value is 2.04%. The network parameters and model size have been greatly reduced by 14.65 M and 55.88 MB, respectively, and the FPS has dropped slightly. Through experiments, it is found that replacing the standard

convolution with the depthwise separable convolution in this study improves the model's accuracy to varying degrees and greatly reduces the number of parameters, making the model more lightweight.

In the experiment, after readjusting the priori boxes with k-means clustering, the model has a certain improvement in various indicators. However, the recall rate is not significantly improved. We consider that there may be insufficient attention to the target object due to too much redundant information in the model, so we add the attention mechanism module in the next experiment. After adding the CBAM attention mechanism to the model, compared with the model without the attention mechanism, the recall rate increased by 2.20%, the mAP increased by 0.12%, the F1 value increased by 0.01, and the precision decreased by 1.03%. The network parameters only increased by 0.021 M, the model size remained almost the same, and the FPS decreased by 1.95. Compared with the model without k-means clustering and attention mechanism module, the recall rate is increased by 4%, the mAP is increased by 2.1%, the F1 value is increased by 0.03, and the precision is increased by 1.56%. From the experimental results, we can see that adding the attention mechanism module greatly improves the model's recall rate while ensuring the model's size and the number of network parameters. The model pays more attention to the characteristic information of maize seedlings.

In general, compared with the YOLOv4 network, when IOU = 0.5, the model proposed in this study can improve mAP by 8.68%, recall by 19.96%, F1 value by 0.13, precision by 5.38%, and reduce the number of network parameters by 70.61%, the model size is reduced by 172.21 MB, and the FPS is increased by 3.71.

The confidence threshold was set as 0.3, so the network could only detect seedling targets higher than the confidence. The recognition effect of YOLOv4, improved YOLOv4 lightweight networks, Mobilenetv1-YOLOv4, Mobilenetv3-YOLOv4, Densenet121-YOLOv4, Vgg-YOLOv4. and on small maize seedlings was compared, as shown in Figure 10. As can be seen from the comparison of areas enclosed by dashed lines in Figure 10a,b, YOLOv4 had misdetection and missed detection of small seedlings, while the improved YOLOv4 model had a better prediction effect.

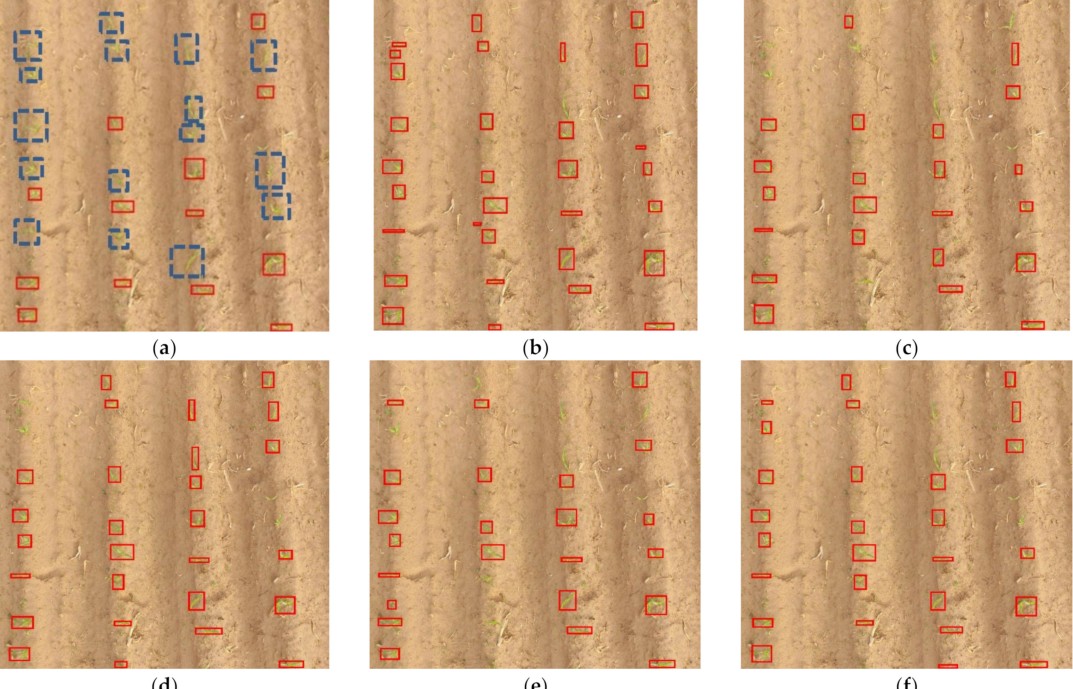

**Figure 10.** Detection results of different models: (**a**) YOLOv4 test results; (**b**) Improved YOLOv4 lightweight networks test results; (**c**) Mbilenetv1-YOLOv4 test results; (**d**) Mobilenetv3-YOLOv4 test results; (**e**) Densenet121-YOLOv4 test results; (**f**) Vgg-YOLOv4 test results.

## 4. Discussion

Many researches have been conducted on the application of object detection in agriculture, such as crop identification, disease detection and plant counting, among which there are some lightweight models with sound effects [43,44]. In this study, the improved YOLOv4 lightweight neural network can obtain more accurate information on the number of maize seedlings to reduce farmers' workload further.

An essential goal of this research is to achieve the model's being lightweight while ensuring detection accuracy. In the maize seedling detection, only one target in the environment needs to be determined, and there is no need to learn too many shape and color features. Therefore, in this study, the backbone network CSPDarknet53 of YOLOv4 is replaced by the improved lightweight network Ghostnet, and the depthwise separable convolution replaces the traditional convolution with fewer parameters. According to the experimental results in Table 3, compared with the 3.77% increase in mAP value when Chenxi Zhang et al. [45] used the improved Ghostnet and introduced the deeply separable convolution model to detect apples, the mAP value of this model is increased by more percentage points, reaching 6.58%. The precision, recall, and F1 value are greatly improved, and the model size is significantly reduced. On the one hand, the improvement of mAP is due to the addition of $1 \times 1$ standard convolution in the Ghost module, which helps improve the ability of the model to provide detailed information. On the other hand, the YOLOV4 model has more complex parameters, more suitable for detecting objects with more categories. However, the target object of this study is only maize seedlings so that the lightweight network will have a better detection effect.

The k-means algorithm is widely used in object detection. Junfeng Gao et al. [46] proposed an improved YOLOv3 model for sugar beet detection. The size of the YOLO anchor box was calculated from the training dataset using the k-means clustering method, and the results showed that the mAP of sugar beet reached 0.897. Since maize seedlings in the data set of this study were at the same growth stage, the size of seedling plants was relatively uniform and consistent. The original prior box is based on 80 categories of detection targets, which is not suitable for single target detection. Therefore, the k-means algorithm is used to optimize the generation of the prior box. The experimental results show that the mAP value of the model using an optimized preselection box is 96.91%, and the precision reaches the maximum value of 97.28%. This is because in multi-scale training, the anchoring framework using the k-means algorithm can better fit the training target and has better generalization ability. In addition, by analyzing the k-means clustering results, it is found that the size of the prior box is relatively concentrated, and most of the targets are small. Therefore, by removing the detection branch that detects large objects. The experimental results show that mAP is improved by 2.4%, and FPS is improved by 12.73 compared with the original model. This shows that the $13 \times 13$ feature layer is unsuitable for small target detection, and the effect is better and the detection speed is faster after removing it.

The model in this study performs well in terms of accuracy, but needs to be improved in terms of FPS. As can be seen from Table 3, the FPS decline of the model is mainly concentrated in the backbone network replacement and the CBAM [47] attention mechanism. In order to solve this problem, a lighter backbone feature extraction network and CBAM module can be built in the subsequent research and embedded in the YOLOv4 network. This can improve the detection accuracy while reducing the weight of the model. In addition, through testing, it was found that the main reasons for false detection and missing detection in this paper are that some weeds and straw were similar in color and shape to maize seedlings, while some seedlings were small in size and not significantly different from the ground. In future studies, we will be more inclined to distinguish the characteristics of weeds and seedlings to improve the detection accuracy of the model further.

## 5. Conclusions

(1) This study proposes an improved YOLOv4 lightweight neural network algorithm for detecting maize seedlings. We used the improved Ghostnet as the backbone feature extraction network to construct the YOLOv4 lightweight network, improved the multi-scale feature fusion network structure, introduced the k-means clustering algorithm to adjust the target prior box, and added the attention mechanism to the neck network to make it more suitable for seedling detection. By introducing deep separable convolution in PANet and YOLO Head networks instead of traditional convolution, the network is more lightweight and more conducive to deploying mobile terminals. The model's training speed and average accuracy are improved by loading the pre-trained weights and freezing some layers.

(2) We verify the feasibility and superiority of the proposed method through comparative experiments on the same test set, taking the F1, recall, mAP, precision, number of model parameters, model size, and FPS as the judgment basis. The method F1, recall rate, mAP, and precision rate of this study are 0.95, 94.02%, 97.03%, and 96.25%, respectively, which are 0.13, 19.96%, 8.68%, and 5.38% higher than YOLOv4. The model network parameters are 18.793 M, the model size is 71.69 MB, and the FPS is 22.92. Compared with the YOLOv4 model, the network parameters are reduced by 70.61%, the model size is reduced by 172.21 MB, and the FPS is increased by 3.71. Through comparative experiments, the model in this study has stronger detection performance, a better prediction effect, and lower model complexity, which is suitable for deployment in edge devices and has a certain application value.

**Author Contributions:** J.G. and J.C. conceived the study and designed the project. J.G. and F.T. performed the experiment, analyzed the data, and drafted the manuscript. B.M. helped to revise the manuscript. All authors have read and agreed to the published version of the manuscript.

**Funding:** This study was funded by the Natural Science Fund Key Project of Heilongjiang Province (ZD2019F002), Heilongjiang Bayi Agricultural University Support Program for San Heng San Zong (ZRCPY202120), the Scientific Research Project of Heilongjiang Provincial Scientific Research Institutes of China (2021YYYF011), and 2020 Daqing City Directive Science and Technology Project (zd-2020-68).

**Institutional Review Board Statement:** Not applicable.

**Informed Consent Statement:** Informed consent was obtained from all subjects involved in the study.

**Data Availability Statement:** Not applicable.

**Conflicts of Interest:** The authors hereby declare that there are no conflicts of interest in the present study.

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
