# Peer review of "A Method for Obtaining the Number of Maize Seedlings Based on the Improved YOLOv4 Lightweight Neural Network"

_agriculture, doi:10.3390/agriculture12101679_

Round 1

Reviewer 1 Report

- Figure 1 could be improved to detect the location of the study area, as the current figure is not representative of the location of the study area.

- The UAV has a CMOS image sensor and a FOV94°20mm f/2.8 lens; please include sensor specifications in the manuscript (spectral and radiometric).

- Please identify all abbreviations at the very least in the first instance.

- When is the best time to detect maize after it has been harvested? When using UAVs, please detect the maize age.

Reviewer 2 Report

Obtaining the number of plants is the key to evaluating the effect of maize mechanical sowing, and is also a reference for subsequent statistics on the number of missing seedlings. This study tries to proposes an improved YOLOv4 lightweight neural network to detect the number of maize seedlings. Although the topic is very interesting, it suffers from major limitations. There are some issues that should be addressed.

Specific comments:

1. The manuscript lacks a condensation of innovation. The proposed method has certain advantages in detection accuracy. However, many similar researches have been published, even with more powerful and rich functions. At the same time, the algorithms and models used in this research are more mature.

2. The method of image acquisition is not clearly stated in the manuscript , and why does the UAV collect images at a height of 10m above the ground? The lower the height above the ground, the clearer the image will be.

3. How was Figure 1 obtained? not stated in the manuscript.

4. A total of 300 images of maize seedlings were collected in this experiment. The amount of data is small, whether it can truly reflect the ability of the model. “The slice images were screened to eliminate fuzzy and distorted images, and 500 images were randomly selected as data sets”, How the slice images were obtained.

5. The formulas and charts in the manuscript are not in standard format, and the authors are advised to revise. Equation 7 is wrong.

6. I did not find Discussion in this manuscript. The authors only list the experimental data but do not analyse them, nor do they analyse and discuss the errors that exist

Round 2

Reviewer 2 Report

Overall the authors have addressed all my questions.